# Effects of a Single Bout of Endurance Exercise on Brain-Derived Neurotrophic Factor in Humans: A Systematic Review and Meta-Analysis of Randomized Controlled Trials

**DOI:** 10.3390/biology12010126

**Published:** 2023-01-13

**Authors:** Zhiqiang Liang, Zheng Zhang, Shuo Qi, Jinglun Yu, Zhen Wei

**Affiliations:** 1Key Laboratory of Exercise and Health Sciences of Ministry of Education, Shanghai University of Sport, Shanghai 200438, China; 2School of Exercise and Health, Shanghai University of Sport, Shanghai 200438, China

**Keywords:** endurance exercise, one session, BDNF, peripheral blood, brain health

## Abstract

**Simple Summary:**

The brain-derived neurotrophic factor (BDNF) is important in mediating the brain health, morphology, and function. A single bout of endurance exercise is a critical factor triggering the expression of the BDNF in peripheral blood. Therefore, investigating the concentration of BDNF in peripheral blood can effectively assess the impact of a single bout of endurance exercise on the brain health. We investigate the impact of a single bout of endurance exercise on the BDNF in humans, meanwhile analyzing how a single bout of endurance exercise impacts the peripheral BDNF types by age group. We found that a single bout of endurance exercise has a small to moderate effect on the BDNF expression. The serum BDNF is a sensitive factor reflecting the effects of a single bout of endurance exercise, regardless of age. These results provide significant evidence for the design of sports exercise to maintain brain health in different age groups.

**Abstract:**

We aimed to investigate the impact of a single bout of endurance exercise on the brain-derived neurotrophic factor (BDNF) in humans and analyze how a single bout of endurance exercise impacts the peripheral BDNF types by age group. We performed a systematic literature review by searching PubMed, Elsevier, and Web of Science for studies that included a single bout of endurance exercise in the experimental group and other exercise types in the control group. Eight interventions were included in the study. Overall, a single bout of endurance exercise significantly increased BDNF expression (SMD = 0.30; 95% CI = [0.08, 0.52]; *p* = 0.001), which was confirmed in the serum BDNF (SMD = 0.30; 95% CI = [0.04, 0.55]; *p* < 0.001). A non-significant trend was observed in the plasma BDNF (SMD = 0.31; 95% CI = [−0.13, 0.76]; *p* = 0.017). The serum and plasma BDNF levels significantly increased regardless of age (SMD = 0.35; 95% CI = [0.11, 0.58]; *p* = 0.004; *I*^2^ = 0%). In conclusion, a single bout of endurance exercise significantly elevates BDNF levels in humans without neurological disorders, regardless of age. The serum BDNF is a more sensitive index than the plasma BDNF in evaluating the impact of a single bout of endurance exercise on the BDNF.

## 1. Introduction

Evidence suggests that endurance exercise can influence brain morphology and function at different ages [1], effectively promoting brain health and preventing age-related loss of cognitive function [2,3]. One of the reasons for these positive effects is that endurance exercise increases the expression of the brain-derived neurotrophic factor (BDNF) [4], which further alters the brain structure and function [2]. The BDNF is a 27.0 kDa homodimer neurotrophic factor comprising two 13.5 kDa subunits joined by noncovalent interactions [5]. The BDNF is produced in the central nervous system and peripheral tissues, stored in the platelets, and easily measured in the peripheral blood [6]. Like most neurotrophies, the BDNF mediates the neurobiology, neuroplasticity, and energy metabolism in the central and peripheral systems [2,7]. Recent data indicates that the BDNF is essential for brain plasticity, neuro-system disease, and cognition capacity [8]. Thus, changing the BDNF through different interventions could positively affect the brain’s neuronal health and cognition, and prevent neurodegenerative diseases [2,9,10].

The BDNF mainly acts by the tyrosine kinase receptor of TrkB to promote neuronal and brain functions [1]. Unlike other neurotrophic factors, such as growth factors, neurotrophine-3, and neurotrophine-4, the BDNF is susceptible to regulating these processes through physical exercise [1,3,7]. Studies suggest that endurance exercise can induce the optimized effects on the BDNF expression and release into the blood by influencing PGC-1α and FNDC5, two crucial metabolic mediators in the brain [11], then other exercise types [12,13]. However, a single bout of endurance exercise (acute exercise training) is a key to triggering these positive processes for the BDNF [1,2]. Therefore, to understand the impact of endurance exercise on the BDNF expression, a single bout of endurance exercise may be an appropriate exercise type compared with other exercise types.

Studies have reported that a single bout of endurance exercise significantly elevates the BDNF concentration in various brain areas [6,10]. Seifert et al. [6] found that a single bout of endurance exercise significantly increased the expression and release of the BDNF in the hippocampus. In addition, recent studies show that an individual’s health status and specific age do not influence the beneficial effects of a single bout of endurance exercise on the BDNF [14]. Thus, the BDNF released from endurance exercise is an ideal way to promote brain health and plasticity among people of different ages [10]. However, these studies combined randomized controlled trials (RCTs) and non-RCTs, which might result in a bias in methodology, influencing the study’s quality. Therefore, further research is needed to investigate the effects of a single bout of endurance exercise on the BDNF.

The BDNF in plasma and serum have been considered indicators representing the BDNF levels in the central nervous system, which positively correlated with performance [15]. Studies used the BDNF in plasma and serum to assess the potential impact of exercise on brain function [16]. For instance, Rodziewicz et al. (2020) measured the BDNF in plasma to investigate the effect of endurance exercise on the BDNF expression [17]. Whereas Arazi et al. [9] measured the BDNF in serum to investigate the effect of endurance exercise on BDNF expression. However, it is unclear if the BDNF in the plasma or the blood serum is better in assessing the effects of a single bout of endurance exercise on the BDNF expression [9]. In addition, Arosio et al. [18] reported that there is a 200-fold difference that exists between serum and plasma BDNF concentrations. Therefore, further studies are needed to confirm the effect of the efficiency of a single bout of endurance exercise in elevating serum and plasma BDNF concentrations.

The primary aim of this study is to investigate the effects of a single bout of endurance exercise on the BDNF expression. We also aim to determine the extent to which the BDNF measured in plasma or serum is more sensitive following a single bout of endurance exercise. We designed a systematic review and meta-analysis to investigate the study’s aims in young, middle, and older-age adults without neurological diseases. 

## 2. Methods

The Preferred Reporting Items for Systematic Reviews and Meta-Analysis (PRISMA 2020) was used to guide the systematic review [19].

### 2.1. Eligibility Criteria and Study Selection

We included studies in the present systematic review and meta-analysis that adhered to the following criteria: (1) they were published in a peer-reviewed journal with full-text availability; (2) the language was English; (3) the study design was a RCT that used either a cross-over or parallel study design and only focused on human studies; (4) a single bout of endurance exercise was considered a whole-body and single-joint exercise, the exercise type was 1 session, and the exercise duration lasted more than 75 s; (5) the participants were adults without neurological disorders.

### 2.2. Search Strategy

The present systematic review was conducted within the PubMed, Elsevier, and Web of Science databases from January 2000 to June 2022. The search strategy consisted of two strategies. The first strategy focused on “endurance exercise” (i.e., “continuous endurance” OR “high-intensity interval exercise” OR “time to task failure” OR “time to exhaustion”). The second strategy focused on the brain-derived neurotrophic factor (i.e., “brain-derived neurotrophic factor” OR “bdnf” OR “BDNF”). The 3 databases identified 335 articles. Before the screening, 39 articles were removed (38 with duplicate records and 1 written in a language other than English). We screened 296 articles for the eligibility criteria, eliminating an additional 273 articles (93%) without a human trial. Of the remaining 23, 15 (65%) were removed following an additional review of the eligibility criteria. Eight remaining articles were included in the systematic review and meta-analysis (Figure 1).

### 2.3. Study Variables and Data Extraction

We examined the effects of a single bout of endurance exercise on the BDNF. The exercise data included different types of a single bout of endurance exercise, including one or more of the following: running, walking, cycling, badminton, and swimming (Table 1). The outcome measures included the serum and plasma BDNF. The means and standard deviations (±SD) were used to compute the effect sizes obtained from the results section, tables, or figures of the reviewed articles. If data were unavailable in a table, we estimated the means ± SD from the published figures using the WebPlotDigitizer software (v4.2, San Francisco, CA, USA).

### 2.4. Risk of Bias Assessment

We used the Cochrane collaboration tool to assess the risk of bias in the RCTs included in the present study. The risk, in terms of selection, performance, detection, attrition, reporting, and others, was judged for each study and classified as high, low, or unclear. The results are presented in Figure 2. The methodology quality for each study was judged by the PEDro scale, ranging from 1 to 10 points. The methodology quality was classified as “poor (≤3 points) ”, “fair (4 to 5 points) ”, “good (6 to 8 points) ”, and “excellent (9 to 10 points)” (Table 2) [22].

### 2.5. Quantitative Analysis

The meta-analysis and statistical analysis were performed in the Review Manager software (RevMan 5.4.1; Cochrane Collaboration, Oxford, UK). The effects of a single bout of endurance exercise on the BNDF were calculated as the differences in the serum and plasma BDNF following exercise between each trial’s experimental and control conditions. Each trial’s standardized mean difference (SMD) values were pooled with a random effects model used to calculate the cumulative effect size and 95% confidence interval. The effect sizes were classified as small (≤0.2), moderate (≤0.5), large (≤0.8). and very large (>0.8). The corresponding z-statistic and p-value were used to investigate whether the cumulative effect size differed significantly from zero. The total heterogeneity was tested using *I*^2^ statistics. A *p*-value < 0.05 was statistically significant.

## 3. Results

### 3.1. Study Selection and Characteristics

The article selection process resulted in 8 interventions, and part of the studies included adults with pathological states, such as multiple sclerosis (MS), obese condition, major depression, and type-2 diabetes. Studies by Ferris et al. [1], Erickson et al. [10], Arazi et al. [9], Behrendt et al. [12], Seifert et al. [6], Schiffer et al. [13], and Zlibinaite et al. [21] studied the effects of a single bout of endurance exercise of periodicity sports on the BDNF in healthy young and older adult groups. Bansi et al. [22], who included subjects with MS, studied the effects of a single bout of endurance exercise training on land and aquatic cycling on the BDNF in middle-aged adults. Studies were published between 2006 and 2021. Table 1 shows the participant and exercise characteristics of the studies.

### 3.2. Study Quality Assessment and Publication Evaluation

The results from the PEDro scale’s bias assessments for the studies are presented in Table 2. The mean score was 6.38 ± 1.59, indicating an overall good quality (Table 2). Five studies [1,6,12,13,21] were of good quality, and three [9,10,20] were of fair quality. The proportion of articles meeting the criteria was as follows: random allocation (100%), concealed allocation (75%), baseline comparability (75%), blind subjects (50%), blind therapists (0%), blind assessors (62.5%), adequate following-up (25%), intention-to-treat analysis (50%), between-group comparison (100%), and point estimates and variability (100%). As for evaluating potential biases, the funnel plots did not indicate significant publication biases for the SMD in this systematic meta-analysis.

### 3.3. Subjects Characters

The RCT studies included 323 subjects (170 in the endurance group; 153 in the control group) (Table 1). For all studies, the gender distribution was 46.43% female, and the average age was 54.41 ± 16.30 (range, 20 to 73 y). Three studies (Ferris et al., Seifert et al., Schiffer et al.) included young adults (mean ages 24.85 ± 4.51 y). Two studies (Zlibinaite et al. and Bansi et al.) included middle-aged adults (mean ages 49.24 ± 7.58 y). Three (Erickson et al., Arazi et al., Behrendt et al.) included older adults (mean ages 65.84 ± 5.57 y). Detailed information on the studies is presented in Table 1.

### 3.4. Characteristics of Endurance Exercise

The single bout of endurance exercise interventions was based on various whole-body periodicity exercises, including cycling, walking, running, rowing, and swimming, and open-skill activities, including badminton. One exercise type was a single bout of a task, with exercises varying from one skill to another, and seven had continuous exercise tasks. The exercise durations ranged from 4 min to 60 min, and the intensity ranged from a 50% maximum heart rate (MHR) to 100% MHR.

### 3.5. Effect of Endurance Exercise on BDNF Expression

Table 3 shows the SMD for the difference between the pre- and post-BDNF values between the endurance exercise and the control groups. The changes in BDNF after the endurance exercise intervention were favored in the mediation of endurance exercise on the neurotrophic BDNF expression. All SMD values were <1.0, with 1 study (Seifert et al.) reaching statistical significance. The CI for the other BDNF values crossed 1.0, suggesting a wide variation. All eight studies averaged were statistically significant (SMD = 0.30; 95% CI = [0.08, 0.52]; *p* = 0.008). Based on the *I*^2^ index (*I*^2^ = 0%), and the studies were homogenous in terms of variability. 

### 3.6. Subgroup Analysis for BDNF in Peripheral Blood

Table 4 shows the SMD for the difference between the BDNF values in serum and BDNF values in plasma between the endurance exercise and the control groups. The subgroup analysis confirmed that endurance exercise significantly benefits the BDNF levels in the serum [1,9,10,12,20]. All SMD values in the serum subgroup were <1.0. The CI for the BDNF values in serum crossed 1.0, suggesting wide variation in values. All five studies in the serum averaged were statistically significant (SMD = 0.30; 95% CI = [0.04, 0.55]; *p* = 0.02). Based on the *I*^2^ index (*I*^2^ = 0%), the studies were homogenous in terms of variability. All SMD values in the plasma subgroup were <1.0, with one study (Seifert et al.) reaching statistical significance. The CI for the BDNF values in plasma crossed 1.0, suggesting wide variation. All three studies averaged were not statically significant (SMD = 0.31; 95% CI = [−0.13, 0.76]; *p* = 0.17) [6,13,21]. The tests for heterogeneity in the plasma subgroup were not statistically significant (χ^2^ = 3.45, *p* = 0.18, *I*^2^ = 42%).

### 3.7. Subgroup Analysis for Subjects

Table 5 shows the SMD for the difference between the age groups between the endurance exercise and the control groups. The subgroup analysis by age groups confirmed the significant benefits of endurance exercise on the BDNF expression in the participants (SMD = 0.35; 95% CI = [0.11, 0.58]; *p* = 0.004; *I*^2^ = 0%). No significant trends were observed in the three groups, young adults (SMD = 0.59; 95% CI = [−0.01, 1.18]; *p* = 0.06; *I*^2^ = 17%) (n = 3 studies) [1,6,13]; middle-age adults (SMD = 0.26; 95% CI = [−0.17, 0.69]; *p* = 0.23; *I*^2^ = 0%) (n = 2 studies) [20,21]; and older adults (SMD = 0.31; 95% CI = [−0.02, 0.65]; *p* = 0.12; *I*^2^ = 0%) (n = 3 studies) [9,10,12]. 

## 4. Discussion

This meta-analysis aimed to investigate the existing evidence from RCT studies on the effect of a single bout endurance exercise on the BDNF (i.e., the serum and plasma BDNF). The results of the 8 endurance exercise studies, which comprised 323 participants, showed a significant effect on the BDNF (SMD = 0.30, *p* = 0.0001), which could be confirmed separately in the serum BDNF (*p* < 0.001). 

The present meta-analysis demonstrated a small to moderate effect of a single bout of endurance exercise on the BDNF (SMD = 0.30). Six of the eight interventions reported the beneficial effects of a single bout of endurance exercise on the BDNF. Prior studies have successfully assessed the beneficial effects of a single bout of endurance on the BDNF in older adults [23], Parkinson’s disease [24], and the chronic post-stroke phase [25]. Systematic reviews and meta-analyses have investigated the effects of various exercise types (i.e., endurance, strength, open skill technique) on the BDNF in the peripheral blood. Marinus et al. [26] and Feter et al. [27] concluded that exercise increases the BDNF levels in adults regardless of age and health status. The recent meta-analysis by Ruiz-Gonzalez et al. [28] reported that different exercise types (i.e., aerobic, resistance, and combined exercise) could significantly increase the BDNF levels in patients with neurodegenerative disorders. Mackay et al. [29] found that endurance exercise can also significantly increase the BDNF levels in patients with neurodegenerative diseases. Marinus et al. [26] revealed that different exercise types, including a single bout of endurance exercise, can increase the BDNF levels in older adults. Our results aligned with previous meta-analyses on the effects of exercise on the BDNF in humans and indicated that endurance exercise could effectively elevate the BDNF levels in different age groups. In addition, we also found that different exercise types show homogeneity in inducing the BDNF expression after a single bout of endurance exercise. Accordingly, differences between exercise types might not be pivotal in influencing the BDNF expression. Zolzaz et al. [16] suggest that the exercise tasks can induce elevating the BDNF expression [16], whereas this induced magnitude of the BDNF expression is correlated with, and depends on, the sport’s intensity [1,16]. In this study, the sport’s intensity ranged from 50% MHR to 75% MHR in all but 1 of the studies. The analogical sport’s intensity among the different exercises caused a similar stimulation on the human body and induced similar performance of the BDNF expression. Thus, despite the different exercises used to assess a one-session endurance exercise on the BDNF expression, the analogical sport’s intensity resulted in the homogeneity of the BDNF expression when integrating results from different studies.

Studies suggests that the BDNF expression is positively associated with synaptic plasticity, neurogenesis, neuroprotection, and cognitive function [27,30], and the elevated BDNF levels after endurance exercise can induce the plasticity of brain morphology and changes in neurophysiology [31]. Erickson et al. [10] reported that the elevated BDNF levels after endurance exercise significantly increased hippocampal volume by 2%, effectively reversing age-related brain volume loss by 1 to 2 years. Ferris et al. [1] also reported that the BDNF is associated with cognitive function. Our findings support the positive benefits of endurance exercise on the elevated BDNF levels. These results corroborate previous evidence supporting the benefits of an endurance exercise intervention on brain health. Therefore, with the positive effects of BDNF on brain health, we recommend that endurance exercise should be an excellent way to maintain neurological health.

We also assessed the effect of a single bout of endurance exercise on different BDNF types in the peripheral blood and observed that a single bout of endurance exercise effectively increases the serum BDNF levels. Accordingly, the serum and plasma BDNF levels reflect the influence of a single bout or chronic exercise on brain health [26,32]. However, no sub-analyses on the difference between the serum and plasma BDNF types could be performed separately for a single bout of endurance exercise in humans. This inability to perform sub-analysis can induce potential bias related to the heterogeneity of different types of a single bout of endurance exercises in terms of their effectiveness. For example, Marinus et al. [26], Feter et al. [27], and Hirsch et al. [31] combined the serum and plasma BDNF levels as critical parameters in their meta-analyses, making sub-analysis impossible. Their results found different beneficial effects between exercise types; sports intensity, exercise session and duration, and exercise modality contribute to heterogeneity in terms of the entire population. The meta-analysis by Ruiz-Gonzalez et al. [28] showed that any exercise could promote the BDNF expression, opposite the results of other studies. Although, they only used the plasma BDNF to monitor the effects of endurance, strength, and combined exercise. 

Physiological studies have shown that the serum and plasma BDNF levels are stored mainly in platelets; however, a 200-fold difference exists between the serum and plasma BDNF levels because the BNDF is released mainly during coagulation [18]. Thus, the difference in physiological composition between the serum and plasma BNDF may result in the contradictions above. The serial serum BDNF generally showed a significant effect size, followed by a single bout of endurance exercise. In addition, the serum BDNF is more sensitive than the plasma BDNF as a parameter in determining the effects of endurance exercise. Thus, the serum BDNF should be used to assess the effects of a single bout of endurance exercise on the BDNF to avoid contradictory findings.

Our sub-analyses also revealed no significant difference in age groups regarding the effects of a single bout of endurance exercise on the BDNF levels. Previous studies suggested that exercise training promotes BDNF levels regardless of health status and age group [26,27]. Our findings not only support the findings of Marius et al. [26] and Feter et al. [27] but also suggest that a single bout of endurance exercise increases the BDNF levels regardless of age group. It is well-documented that physical exercise can stimulate the BDNF expression and increase the BDNF concentration in humans. This physical training-induced up-regulation of the BDNF expression plays a vital role in human health. The BDNF can improve human functioning capacities, such as strength and endurance performance, mood, and cognitive functions [16]. The study even reported that the effects of exercise on the BDNF produce the similar effects of pharmacological treatment with antidepressant drug [16]. Therefore, the findings from our study provide insights for clinical and rehabilitation practitioners to design the prescription of exercise intervention for neurological disorders.

This systematic review and meta-analysis’s strengths are in evaluating RCT-derived evidence on the effects of a single bout of endurance exercise on the BDNF expression in the peripheral blood in humans without neurological disorders and in analyzing which BDNF type is more sensitive to the beneficial effects of a single bout of endurance exercise. Moreover, unlike previous systematic reviews and meta-analyses, the present study only evaluated RCTs. However, the systematic review and meta-analysis have some limitations. First, most of the included RCTs analyzed various closed-loop endurance exercise types. Only one study analyzed open-skill exercises. Other exercise types, such as endurance strength training and functional threshold power endurance exercise, were not analyzed. Therefore, further studies are needed to clarify the effects of a single bout of endurance exercise on BDNF expression in different endurance exercise types. Second, many included studies focused on moderate-intensity sports and rarely investigated the effects of high-intensity exercises. Previous studies suggested that the increased magnitude of the BNDF expression in healthy humans depends on exercise intensity [1]. A positive linear relationship exists between exercise intensity and the effects of a single bout of exercise on the BDNF levels [33]. High-intensity interval training (HIIT) has recently received considerable attention because of its time-efficient exercise characteristics [34]. Emerging evidence suggests that HIIT may elicit superior health benefits than continuous endurance exercise [34]. Accordingly, HIIT is used as a physical method in clinical, rehabilitation, and public health to maintain good health in different populations [35]. However, whether HIT also exerts significantly beneficial effects by elevating the BDNF levels relative to continuous endurance exercise is unknown. Therefore, clarifying the relationship between the BDNF changes and high-intensity exercise is necessary. Given that exercise remains an effective strategy in regulating the BDNF expression, clarifying this relationship maximizes the design of therapeutic strategies to maintain brain health and intervention in aging problems. 

## 5. Conclusions

The present systematic review and meta-analysis provides evidence from RCTs that a single bout of endurance exercise elevates the BDNF expression in humans without limb disabilities. Compared with the plasma BDNF, the serum BDNF is more sensitive to the effects of a single bout of endurance exercise regardless of age. Given the critical role of the BDNF in the brain and neuronal health, a single bout of endurance exercise should be considered as an intervention to maintain brain health in different age groups. Further high-quality research is warranted to investigate the mechanisms of how non-closed-loop endurance exercise, with or without high-intensity exercise, affects the serum BDNF.

## Figures and Tables

**Figure 1 biology-12-00126-f001:**
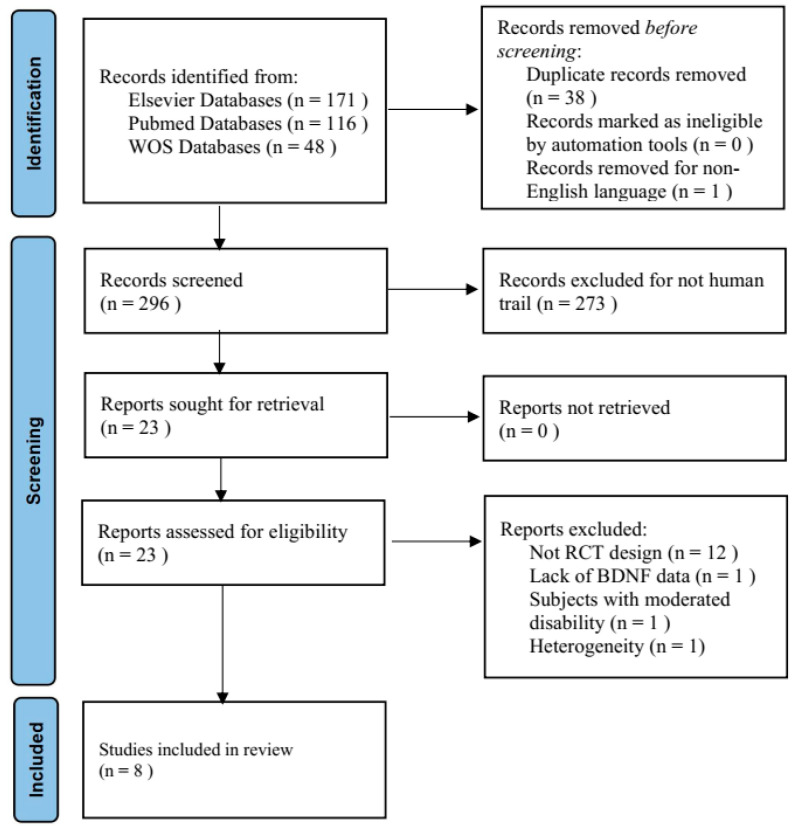
Flow diagram of literature search.

**Figure 2 biology-12-00126-f002:**
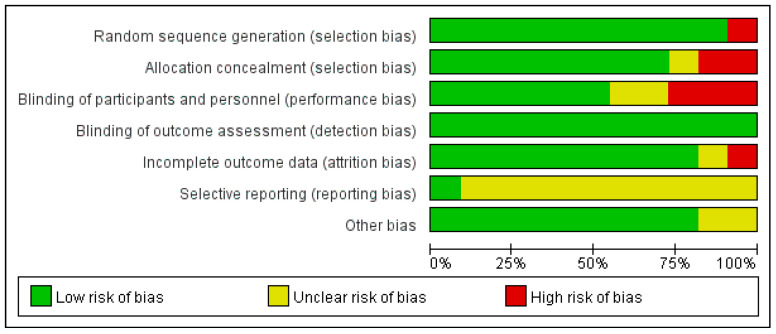
The evaluation of potential biases for included studies.

**Table 1 biology-12-00126-t001:** Participant and exercise characteristics of included studies (N = 323; n = 170 endurance group; n = 153 control group).

Study	Subjects (Female)	Age (y) ± SD	Height (cm) ± SD	Weight (kg) ± SD	BDNF	Volume (mL)	Location	Exercise Type (Treatment)	Time (min)	Sports Intensity	Main Outcome (BDNF pg/mL) ± SD
Ferris et al. [1]	15(11)	25.4 ± 1.0	174.7 ± 1.9	71.0 ± 3.1	Serum	10	Antecubital vein	Cycling (treatment)	30	Vth plus 10% vs. Vth minus 20%	10%: 21,000 ± 2000)
											20%: 20,000(2000)
Erickson et al. [10]	WE: 60 (44) ST: 60 (36)	67.6 ± 5.81 65.5 ± 5.44	-	-	Serum	-	Antecubital vein	Walk (treatment) vs. Stretching	40	50–6 0% to 60–7 5% MHRR	E: 23.77 ± 8.04
											C: 24.04 ± 10.83
Arazi et al. [9]	E: 10 (0) S: 10 (0)	60.7 ± 1.7 60.8 ± 1.8	181 ± 5.4 177.8 ± 8.9	85.9 ± 13.4 90.6 ± 16	Serum	10	Antecubital vein	Running (treatment) vs. Resistance training	30	65–70% MHR	E: 4.68 ± 3.9
											C: 3.21 ± 0.29
Behrendt et al. [12]	AG: 24 (12) CG: 14 (6)	65.83 ± 5.98 67.07 ± 2.37	173 ± 10 168 ± 8	84.05 ± 16.51 76.84 ± 12.69	Plasma	-	Median cubital vein or Cephalic vein	Badminton (OSE) Bicycling Control	30	60 ± 5% HRR	aOSE: 1894.63 ± 780
											aCSE: 1682.04 ± 807.15
											CG: 1950.14 ± 1043.07
Seifert et al. [6]	E: 7 (0) CG: 5 (0)	29 ± 6.0 31 ± 7.0	181 ± 6 184 ± 7	90.1 ± 7.7 96.2 ± 7.7	Plasma	-	Arterial and internal jugular venous	Cycling, running, swimming, and rowing (treatment) vs. Sedentary lifestyle	60	70% of MHR or 65% of VO2max	E: 4.4 ± 2.4
											C: 1.3 ± 0.3
Schiffer et al. [13]	E: 9 (0) S: 9 (0) CG: 9 (0)	23 ± 1.7 22 ± 1.6 22 ± 2.3	184 ± 2.7 183 ± 6.6 183 ± 7.5	82 ± 8.6 77 ± 6.8 75 ± 8.2	Plasma	-	Cubital vein	Running vs. Strength training vs. Control group	45	80% of HR	E: 128.4 ± 90.2
											S: 136 ± 109
											C: 102.2 ± 108.7
Zlibinaite et al. [20]	E: 17 (0)	44.8 ± 6.5		94.3 ± 12.3	Serum	5	Vein	Graded increased cycling exercise	4–1 3	100% of MHR	E: 1500 ± 200
	CG: 16 (0)	48.8 ± 5.3		91.3 ± 19.9							C: 1200 ± 900
Bansi et al. [21]	ELG: 28 (13) EWG: 24 (17)	51.64 ± 7.49 49.89 ± 8.27	166.14 ± 3.4 169.57 ± 4.7	69.11 ± 6.95 69.36 ± 9.61	Serum	-	Antecubital vein	Cycling vs. Aquatic cycling	30	70% of MHR or 60% of VO2max	E: 17,138.13 ± 7284.54
											C: 16,520.1 ± 6326.37
											C: 1200 ± 900
**Age group**											
**Young adults**	60 (11)	24.85 ± 4.51	180.69 ± 6.53	79.81 ± 10.81							
**Experimental group**	31 (11)	25.52 ± 3.75	178.82 ± 5.39	78.51 ± 10.03							
**Control group**	29 (11)	25.31 ± 4.43	178.88 ± 6.83	76.89 ± 10.92							
**Middle adults**	85 (30)	49.24 ± 7.58	167.72 ± 4.40	78.40 ± 16.74							
**Experimental group**	45 (13)	49.05 ± 7.87	166.14 ± 3.4	78.63 ± 15.37							
**Control group**	40 (17)	49.45 ± 7.25	169.57 ± 4.7	78.14 ± 8.15							
**Older adults**	178 (98)	65.84 ± 5.57	174 ± 9.78	83.76 ± 15.72							
**Experimental group**	94 (56)	66.41 ± 5.95	175.35 ± 9.62	84.59 ± 15.68							
**Control group**	84 (42)	65.20 ± 5.04	172.08 ± 9.68	82.57 ± 15.7							
**Group**											
**Experimental group**	170 (80)	54.36 ± 16.65	173.55 ± 9.00	81.90 ± 15.03							
**Control group**	153 (70)	40.52 ± 16.17	172.71 ± 8.72	80.33 ± 13.47							
**Overall**	323 (150)	54.41 ± 16.30	173.46 ± 9.17	81.65 ± 15.48							

Definitions: Vth: ventilatory threshold; WE: endurance walking; ST: stretching; MHRR: maximum heart rate reserve; E: endurance; S: strength; AG: aerobic group; MHR: maximal heart rate; aOSE: open skill exercise; aCSE: closed skill exercise; CG: control group; HRR: heart rate reserve; Con: continue cycling exercise; ELG: ergometer land group; EWG: ergometer water group.

**Table 2 biology-12-00126-t002:** Methodological quality of studies included systematic review.

Author Year	Random Allocation	Concealed Allocation	Baseline Comparability	Blind Subjects	Blind Therapists	Blind Assessors	Adequate Following-Up	Intention-to-Treat Analysis	Between-Group Comparison	Point Estimates and Variability	Total (0–10 Score)
**Young adults**											
Ferris et al. [1]	Yes	No	No	Yes	No	Yes	No	Yes	Yes	Yes	6
Schiffer et al. [13]	Yes	Yes	Yes	Yes	No	Yes	No	Yes	Yes	Yes	8
Seifert et al. [6]	Yes	Yes	Yes	Yes	No	Yes	No	Yes	Yes	Yes	8
**Middle adults**											
Bansi et al. [21]	Yes	Yes	Yes	No	No	Yes	Yes	No	Yes	Yes	7
Zlibinaite et al. [20]	Yes	No	Yes	No	No	No	Yes	No	Yes	Yes	5
**Older adults**											
Erickson et al. [10]	Yes	Yes	No	No	No	No	No	No	Yes	Yes	4
Arazi et al. [9]	Yes	Yes	Yes	No	No	No	No	No	Yes	Yes	5
Behrendt et al. [12]	Yes	Yes	Yes	Yes	No	Yes	No	Yes	Yes	Yes	8

**Table 3 biology-12-00126-t003:** Standardized mean differences and forest plot of comparison of endurance exercise group and control group on BDNF.

		Experimental			Control				Std. Mean Difference		
Study or Subgroup		Mean	SD	Total	Mean	SD	Total	Weight	IV, Random, 95% CI		
Arazi et al. [9]		4.68	3.9	10	3.21	0.29	10	6.20%	0.51	[−0.38, 1.40]	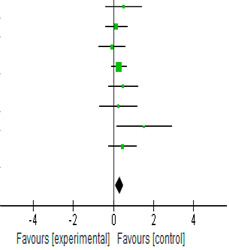
Bansi et al. [21] 2012		17,138.1	4662.6	28	16,520.1	44,013.9	24	16.50%	0.14	[−0.41, 0.69]	
Behrendt et al. [12]		1894.63	780	24	19,540.14	1043.07	14	11.30%	−0.06	[−0.72, 0.60]	
Erickson et al. [10]		23.77	8.04	60	21.32	9.32	60	38.10%	0.28	[−0.08, 0.64]	
Ferris et al. [1]		21,000	2000	15	20,000	2000	15	9.30%	0.49	[−0.24, 1.21]	
Schiffer et al. [13]		128.4	90.2	9	102.2	108.7	9	5.70%	0.25	[−0.68, 1.18]	
Seifert et al. [6]		4.4	2.4	7	1.3	0.3	5	2.6	1.53	[0.16, 2.90]	
Zlibinaite et al. [20]		1500	200	17	1200	900	16	10.30%	0.46	[−0.24, 1.15]	
Total (95% CI)				170			153	100.00%	0.3	[0.08, 0.52]	
Heterogeneity: Tau^2^ = 0.00; Chi^2^ = 5.27, df = 7 (*p* = 0.63); *I*^2^ = 0%											
Test for overall effect: Z = 2.65 (*p* = 0.008)											

**Table 4 biology-12-00126-t004:** Standardized mean differences and forest plot of impact of endurance exercise on BDNF levels in serum and plasma.

		Experimental			Control				Std.Mean Difference			
Study or Subgroup		Mean	SD	Total	Mean	SD	Total	Weight	IV, Random, 95% CI			
**1.1.1 Serum**												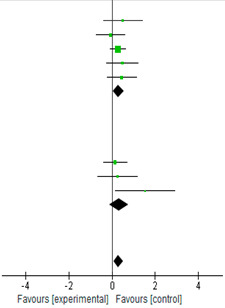
Arazi et al. [9]		4.68	3.9	10	3.21	0.29	10	6.20%	0.51	[−0.38, 1.40]		
Behrendt et al. [12]		1894.63	780	24	19,540.14	1043.07	14	11.30%	−0.06	[−0.72, 0.60]		
Erickson et al. [10]		23.77	8.04	60	21.32	9.32	60	38.10%	0.28	[−0.08, 0.64]		
Ferris et al. [1]		21,000	2000	15	20,000	2000	15	9.30%	0.49	[−0.24, 1.21]		
Zlibinaite et al. [20]		1500	200	17	1200	900	16	10.30%	0.46	[−0.24, 1.15]		
Subtotal (95% CI)				126			115	75.10%	0.3	[0.04, 0.55]		
Heterogeneity: Chi^2^ = 1.82, df = 4 (*p* = 0.77); *I*^2^ = 0%												
Test for overall effect: Z = 2.27 (*p* = 0.02)												
**1.1.2 Plasma**												
Bansi et al. [21]		17,138.1	4662.6	28	16,520.1	44,013.9	24	16.50%	0.14	[−0.41, 0.69]		
Schiffer et al [13]		128.4	90.2	9	102.2	108.7	9	5.70%	0.25	[−0.68, 1.18]		
Seifert et al [6]		4.4	2.4	7	1.3	0.3	5	2.6	1.53	[0.16, 2.90]		
Subtotal (95% CI)				44			38	24,9%	0.31	[−0.13, 0.76]		
Heterogeneity: Chi^2^ = 3.45, df = 2 (*p* = 0.18); *I*^2^ = 42%												
Test for overall effect: Z = 1.37 (*p* = 0.17)												
Total (95% CI)				170			153	100.00%	0.3	[0.08, 0.52]		
Heterogeneity: Chi^2^ = 5.27, df = 7 (*p* = 0.63); *I*^2^ = 0%												
Test for overall effect: Z = 2.65 (*p* = 0.008)												
Test for subgroup differerence: Chi^2^ = 0.00, df = 1 (*p* = 0.95); *I*^2^ = 0%												

**Table 5 biology-12-00126-t005:** Standardized mean differences and forest plot of impact of endurance exercise on BDNF levels in different populations.

		Experimental			Control				Std. Mean Difference			
Study or Subgroup		Mean	SD	Total	Mean	SD	Total	Weight	IV, Random, 95% CI			
**1.2.1 Yong adult**												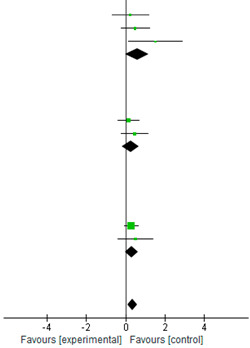
Ferris et al. [1]		21,000	2000	15	20,000	2000	15	9.30%	0.49	[−0.24, 1.21]		
Schiffer et al. [13]		128.4	90.2	9	102.2	108.7	9	5.70%	0.25	[−0.68, 1.18]		
Seifert et al. [6]		4.4	2.4	7	1.3	0.3	5	2.6	1.53	[0.16, 2.90]		
**Subtotal (95% CI)**				31			29	19.90%	0.59	[−0.01, 1.18]		
Heterogeneity: Tau^2^ = 0.05; Chi^2^ = 2.40, df = 2 (*p* = 0.30); *I*^2^ = 17%												
Test for overall effect: Z = 1.93 (*p* = 0.05)												
**1.2.2 Middle adult**												
Bansi et al. [21]		17,138.1	4662.6	28	16,520.1	44,013.9	24	16.50%	0.14	[−0.41, 0.69]		
Zlibinaite et al. [20]		1500	200	17	1200	900	16	10.30%	0.46	[−0.24, 1.15]		
**Subtotal (95% CI)**				45			40	30.20%	0.26	[−0.17, 0.69]		
Heterogeneity: Tau^2^ = 0.00; Chi^2^ = 0.49, df = 1 (*p* = 0.48); *I*^2^ = 0%												
Test for overall effect: Z = 1.19 (*p* = 0.23)												
**1.2.3 Older adult**												
Arazi et al. [9]		4.68	3.9	10	3.21	0.29	10	6.20%	0.51	[−0.38, 1.40]		
Behrendt et al. [12]		1894.63	780	24	19,540.14	1043.07	14	11.30%	−0.06	[−0.72, 0.60]		
Erickson et al. [10]		23.77	8.04	60	21.32	9.32	60	38.10%	0.28	[−0.08, 0.64]		
**Subtotal (95% CI)**				70			70	49.90%	0.31	[−0.02, 0.65]		
Heterogeneity: Tau^2^ = 0.00; Chi^2^ = 0.22, df = 1 (*p* = 0.64); *I*^2^ = 0%												
Test for overall effect: Z = 1.83 (P = 0.07)												
**Total (95% CI)**				170			153	100.00%	0.30	[0.08, 0.52]		
Heterogeneity: Tau^2^ = 0.00; Chi^2^ = 3.97, df = 6 (*p* = 0.68); *I*^2^ = 0%												
Test for overall effect: Z = 2.88 (*p* = 0.008)												
Test for subgroup difference: Chi^2^ = 0.82, df = 2 (*p* = 0.66); *I*^2^ = 0%												

## Data Availability

Data will be available upon request from the corresponding author.

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
