# Peer review of "Effects of a Single Bout of Endurance Exercise on Brain-Derived Neurotrophic Factor in Humans: A Systematic Review and Meta-Analysis of Randomized Controlled Trials"

_biology, 2023, doi:10.3390/biology12010126_

Round 1

Reviewer 1 Report

The study presented by Dr. Liang et al., performs a meta-analysis review on the acute effect of exercise in humans of different ages included in 8 different original articles. They conclude that acute exercise significantly elevates BDNF levels in healthy humans regardless of age, being this difference more evident in serum than in plasma. Overall, this is a well-conducted study, however I have concerns which should be addressed before being suitable for publication.

It would be worthwhile performing an accurate revision of the spelling and grammar through the manuscript. Some examples are listed below:

Line 13: has been confirm should be has been confirmed

Line 15: Lack of space after the point

Line 18: BDN expression should be BDNF expression

Line 45 and throughout: neurotrophies should be neurotrophins or neurotrophic factors

Lines 132-134 should be deleted since they are from the template

Introduction:

Researchers should consider that in neurodegenerative diseases such as Amyotrohpic Lateral Sclerosis (ALS), BDNF levels are increased in comparison with healthy controls and exogenous administration did not induce any beneficial effect. Therefore, analysing BDNF levels alone and relating them with benefits resulting from exercise can lead to misleading interpretations since its action strongly depends on receptor availability. Authors should revise the statements done through the manuscript concerning this subject. Also, it could be of interest to perform the same analyse over TrkB and p75 receptors.

Methods:

Section 2.1: please refer that this work has been using only human studies

Line 89: this statement is confusing since endurance training is classically related with moderate intensity exercise, partially recruiting muscular fibres to avoid reaching muscular fatigue in a short period of time.

Line 102: this statement is confusing since strength training is classically considered as resistance training and not endurance. It would be worthwhile to revise the usage of endurance training classification throughout the manuscript.

Section 2.4 heading is duplicated

Results:

Line 137: this statement should be corrected or removed since Giuseppe Schirò et al., 2022 (doi: 10.3389/fneur.2022.917527) recently reported the impact of the BDNF/TrkB signalling pathway in MS. In the same line, authors should mention that the selected papers to do this meta-analysis included pathological states such as obese condition (Zlibinaite et al., 2010) or major depression and type-2 diabetes (Seifert et al., 2010).

Table 1: All definitions of abbreviations should appear at the end of the table. Some of them like AG or ST are missing. Also, this article would benefit of clearly identifying the experimental conditions of each one of the papers to ensure readers consciousness of the differences included in the meta-analysis.

Section 3.4 would benefit of identifying which one of the articles was using continuous exercise tasks and why it was considered acute exercise. Indeed, Seifert et al., 2010 performed a 3 month training protocol and therefore should not be considered acute exercise and should be removed from this study. Also, this could be the explanation of the strongest increase in BDNF levels found on this paper.

Section 3.5 would benefit of identifying the method used to calculate BDNF abundance and differences in the different studies.

Tables 3-5 present duplicate information. It would be worthwhile to find a better way to show and integrate the information.

Discussion:

The discussion section is repetitive and does not give an explanation about the homogeneity of the results despite of the different participants included in the study and the differences through exercise protocols in the original articles. The beneficial effects of BDNF and its increase with exercise have been already widely described and the discussion of this paper should add a new point of view on the interpretation of the data further than repeating the information already given in the original papers, especially considering that there are other recent meta-analysis reporting the same conclusions.

Authors should avoid statements such as “Our findings support the positive benefits of elevated BDNF levels on brain health after acute endurance exercise. Therefore, acute endurance exercise is vital to maintain neuro-logical health.” as they do not report any benefits but just an increase on its levels, which is subrogated to receptor availability to be effective.

Author Response

Comments

Correction

The study presented by Dr. Liang et al., performs a meta-analysis review on the acute effect of exercise in humans of different ages included in 8 different original articles. They conclude that acute exercise significantly elevates BDNF levels in healthy humans regardless of age, being this difference more evident in serum than in plasma. Overall, this is a well-conducted study, however I have concerns which should be addressed before being suitable for publication.

It would be worthwhile performing an accurate revision of the spelling and grammar through the manuscript. Some examples are listed below:

Thank you for your comments. We have corrected the manuscript for grammar and spelling errors This required us to make some minor changes to some sentences to improve the sentence structure.

Line 13: has been confirm should be has been confirmed

We have corrected and reorganized this sentence from “Acute endurance exercise has been confirm that can effectively trigger increase of the expression of BDNF in peripheral blood. Therefore, investigating impact of acute endurance exercise on BDNF expression in peripheral blood is necessary” to “Whereas, acute endurance exercise is the key factor to trigger the expression of BDNF in peripheral blood. Therefore, investigating concentration of BDNF in peripheral blood can effectively assess impact of acute endurance exercise on brain health”.

Line 15: Lack of space after the point

We have added the space after the point.

Line 18: BDN expression should be BDNF expression

We have corrected this typo from BDN to BDNF.

Line 45 and throughout: neurotrophies should be neurotrophins or neurotrophic factors

We have change this from neurotrophies to neurotrophic factors

Lines 132-134 should be deleted since they are from the template

We have deleted this template.

Introduction:

Researchers should consider that in neurodegenerative diseases such as Amyotrohpic Lateral Sclerosis (ALS), BDNF levels are increased in comparison with healthy controls and exogenous administration did not induce any beneficial effect. Therefore, analysing BDNF levels alone and relating them with benefits resulting from exercise can lead to misleading interpretations since its action strongly depends on receptor availability. Authors should revise the statements done through the manuscript concerning this subject. Also, it could be of interest to perform the same analyse over TrkB and p75 receptors.

We recognize the point and . clearly showing the relationship between BDNF expression and receptor of TrkB and p75 would further understanding of the relationship between BDNF expression and acute endurance exercise. This topic is outside the scope of our study. However, , we plan to study TrkB and p75 receptors in a systematic review and meta-analysis on in a subsequent study.

Methods:

Section 2.1: please refer that this work has been using only human studies

We have added this suggestion into section 2.1.

Please review the point 3 in section 2.1.

Line 89: this statement is confusing since endurance training is classically related with moderate intensity exercise, partially recruiting muscular fibres to avoid reaching muscular fatigue in a short period of time.

 We cited articles in the last paragraph of the discussion showing the benefits of high-intensity interval training (HIIT) in the discussion (i.e., Wewege et al., 2017; Leanna et al. 2016). Studies show that HIIT can induce superior health benefits better than continuous endurance exercise (Leanna et al.) and it deserves further study.   

Line 102: this statement is confusing since strength training is classically considered as resistance training and not endurance. It would be worthwhile to revise the usage of endurance training classification throughout the manuscript.

We have changed our statement to make it clearly, we delete “strength training” from our manuscript.

Section 2.4 heading is duplicated

We have change this heading from “2.4 Risk of Bias Assessment” to “2.5 Quantitative Analysis”.

Results:

Line 137: this statement should be corrected or removed since Giuseppe Schirò et al., 2022 (doi: 10.3389/fneur.2022.917527) recently reported the impact of the BDNF/TrkB signalling pathway in MS. In the same line, authors should mention that the selected papers to do this meta-analysis included pathological states such as obese condition (Zlibinaite et al., 2010) or major depression and type-2 diabetes (Seifert et al., 2010).

We have change these statements from “ The article-selection process resulted in 8 interventions, 7 in healthy adults and one included adults with multiple sclerosis (MS), obese condition, major depression and type-2 diabetes. This immune system disorder that does not interfere with the release of BDNF following exercise.” to “The article-selection process resulted in 8 interventions, part of studies included adults with pathological states, such as multiple sclerosis (MS), obese condition, major depression and type-2 diabetes”.

Table 1: All definitions of abbreviations should appear at the end of the table. Some of them like AG or ST are missing. Also, this article would benefit of clearly identifying the experimental conditions of each one of the papers to ensure readers consciousness of the differences included in the meta-analysis.

We have checked the definition of abbreviations and added the definition of AG at the end of the table and identified which of the exercise groups are the treatment group.

Section 3.4 would benefit of identifying which one of the articles was using continuous exercise tasks and why it was considered acute exercise. Indeed, Seifert et al., 2010 performed a 3 month training protocol and therefore should not be considered acute exercise and should be removed from this study. Also, this could be the explanation of the strongest increase in BDNF levels found on this paper.

In sports science area, “Continuous exercise tasks” is considered as one of exercise types, where humans  constantly repeat the movements until end of exercise. In the present study, we found that the exercise types in studies were running, walking, cycling, rowing and swimming. These exercise types require the human body to repeat a movement for a long time. Therefore, they are a continuous exercise task. As the exercise sessions included  one-session of exercise, we considered this to be one-session of an endurance-type exercise among these studies as acute endurance exercise.

Seifert et al. (2010) indeed performed a 3-month training protocol, which including the baseline and 3-month follow-up exercise test. The baseline test included measured a  one-session endurance exercise bout on BDNF expression. This is categorized as an acute exercise in the sports science area, therefore, we included this study into our meta-analysis.

In addition, we also realized that analyzing the chronic effect of endurance exercise on BDNF is necessary to understand the relationship between endurance exercise on BDNF expression. We are going to take this analysis in a subsequent study.

Section 3.5 would benefit of identifying the method used to calculate BDNF abundance and differences in the different studies.

We checked the paper we include again. These papers clearly show where they collect the BDNF sample. We have carefully considered your suggestions; however, we do not find a appropriate way to figure out this limitation.

Tables 3-5 present duplicate information. It would be worthwhile to find a better way to show and integrate the information.

We have recognizeded the table, and hope it can clearly show and integrate the information about BDNF and acute endurance exercise.

Discussion:

The discussion section is repetitive and does not give an explanation about the homogeneity of the results despite of the different participants included in the study and the differences through exercise protocols in the original articles. The beneficial effects of BDNF and its increase with exercise have been already widely described and the discussion of this paper should add a new point of view on the interpretation of the data further than repeating the information already given in the original papers, especially considering that there are other recent meta-analysis reporting the same conclusions.

We readjusted and the discussion to discuses the homogeneity of results from the exercise type and sports intensity. Because previous study suggest that exercise tasks and sports intensity are correlated with BDNF expression; the outcome of exercise types on the human body can increase blood platelets, and further increasing BDNF expression. This increased magnitude of BDNF expression is also dependent on exercise intensity. Higher exercise intensity might induce higher BDNF expression. Therefore, we inclluded this information according to above research findings. The detailed statement (from line 258 to line 277)is : Removed: This indicates that the difference between exercise types might be not a key factor to influence the BDNF expression. Previous study suggest that the exercise tasks can induce elevating BDNF expression; whereas, this induced magnitude of BDNF expression is correlated with and depended on the sports intensity [1] . In this study, the sports intensity in all but one of studies are range from 50% MHR to 75% MHR. This analogical sports intensity among different exercises causes the similar stimulation on human body and induce similar performance of BDNF expression. Therefore, although different exercises were used to assess one-session endurance exercise on BDNF expression, the analogical sprorts intensity results in the homeogeneity of BDNF expression when intergrate these results from different studies.

Study suggests that BDNF expression has been associated positively with synaptic plasticity, neurogenesis, neuroprotection, and cognitive function [24,27]; elevated BDNF levels after endurance exercise can induce the plasticity of brain morphology and changes in neurophysiology [27]. Erickson et al. [9] reported that elevated BDNF levels after endurance exercise significantly increased hippocampal volume by 2%, effectively reversing age-related brain volume loss by 1–2 years. Ferris et al. [1] also reported that BDNF is associated with cognitive function. Our findings support the positive benefits of endurnce exercise on elevated BDNF levels These results corroborate pervious evidence supporting the benefits of endurance exercise intervention on brain health. Therefore, with the positive effects of BDNF on brain health, we recommod that endurance exercise should be a good way to maintain neurological health.

Authors should avoid statements such as “Our findings support the positive benefits of elevated BDNF levels on brain health after acute endurance exercise. Therefore, acute endurance exercise is vital to maintain neuro-logical health.” as they do not report any benefits but just an increase on its levels, which is subrogated to receptor availability to be effective.

We have changed this statement and reorganize it from “Our findings support the positive benefits of elevated BDNF levels on brain health after acute endurance exercise. Therefore, acute endurance exercise is vital to maintain neuro-logical health. “ to “These results corroborate pervious evidence supporting the benefits of endurance exercise intervention on brain health. Therefore, with the positive effects of BDNF on brain health, we recommod that endurance exercise should be a good way to maintain neurological health.”

Reviewer 2 Report

"Effects of Acute Endurance Exercise on Brain-Derived Neurotrophic Factor in Humans: A Systematic Review and Meta-Analysis of Randomized Controlled Trials" by Liang and colleagues is a systematic literature review and meta-analysis that investigated the impact of acute endurance exercise on brain-derived neurotrophic factor (BDNF) in humans, as well as assessed how acute endurance exercise may affect peripheral BDNF types according to age group.

The topic is very interesting and of current interest in the scientific literature. The manuscript is well-written and well-structured and provides interesting results that could improve and expand current knowledge on the topic. Therefore, I believe that a few minor revisions could greatly improve the manuscript and make it suitable for publication in Biology.

Line 18 - Please correct “BDN” with “BDNF”.

Line 26 - Please replace “brain-derived neurotrophic factor (BDNF)” with “BDNF”, as the full name has already been given in the previous lines.

Introduction - I believe the introduction could be improved by including more recent evidence supporting the positive effects of BDNF on brain health. For example, the authors could refer to a recent publication that explored how different types of exercise differentially modulate cognitive and cerebellar functions (doi: 10.3390/ijms231810388). The authors repeatedly compare the function of BDNF to that of other neurotrophins. It would be appropriate to mention a few of them (doi: 10.1016/j.jshs.2019.07.012). Furthermore, the authors could expand on the information gained from the work of Wrann et al. to explain the potential biological mechanism by which BDNF influences brain function (doi: 10.1016/j.cmet.2013.09.008.).

Author Response

Comments

Correction

"Effects of Acute Endurance Exercise on Brain-Derived Neurotrophic Factor in Humans: A Systematic Review and Meta-Analysis of Randomized Controlled Trials" by Liang and colleagues is a systematic literature review and meta-analysis that investigated the impact of acute endurance exercise on brain-derived neurotrophic factor (BDNF) in humans, as well as assessed how acute endurance exercise may affect peripheral BDNF types according to age group.

The topic is very interesting and of current interest in the scientific literature. The manuscript is well-written and well-structured and provides interesting results that could improve and expand current knowledge on the topic. Therefore, I believe that a few minor revisions could greatly improve the manuscript and make it suitable for publication in Biology.

Thank you for your comments about our study and manuscript.

Line 18 - Please correct “BDN” with “BDNF”.

We have corrected this typo from “BDN” to “BDNF”.

Line 26 - Please replace “brain-derived neurotrophic factor (BDNF)” with “BDNF”, as the full name has already been given in the previous lines.

We have deleted brain-derived neurotrophic factor, and only keep the abbreviation of BDNF according to your comments.

Introduction - I believe the introduction could be improved by including more recent evidence supporting the positive effects of BDNF on brain health. For example, the authors could refer to a recent publication that explored how different types of exercise differentially modulate cognitive and cerebellar functions (doi: 10.3390/ijms231810388). The authors repeatedly compare the function of BDNF to that of other neurotrophins. It would be appropriate to mention a few of them (doi: 10.1016/j.jshs.2019.07.012). -ed from the work of Wrann et al. to explain the potential biological mechanism by which BDNF influences brain function (doi: 10.1016/j.cmet.2013.09.008.)

These suggestions are very useful to further increase understanding on mechanism of BDNF expression. Accordingly, we have revised the introduction part of manuscript and added the references.

Reviewer 3 Report

This meta-analysis reviewed RCTs that investigated the effect of acute endurance exercise on the level of brain-derived neurotrophic factor (BDNF). I have some comments.

Simple summary

1. Line 12: “Brain-derived neurotrophic factor plays an important role in …” Please add the abbreviation, BDNF, for brain-derived neurotrophic factor here.

2. Line 13-15: “Acute endurance exercise has been confirm that can effectively trigger increase of the expression of BDNF in peripheral blood. Therefore, investigating impact of acute endurance exercise on BDNF expression in peripheral blood is necessary.” These two sentences are confusing. Please rewrite them to make it clearer.

Introduction

1. Line 55-56:” Several studies have reported that acute endurance exercise significantly elevates the BDNF concentration in various brain areas.” Please add references here.

2. Please consider rewriting the introduction part. It is unclear to me why the authors focused on the effect of acute endurance exercise instead of other exercises on BDNF.

3. In the discussion part, the authors mentioned the difference between plasma and serum BDNF. Please describe plasma and serum BDNF in the introduction part as well, since this study tested the effects of acute endurance exercise on plasma and serum BDNF separately.

4. The authors aimed to test the effect of acute endurance exercise in different age groups. This part also needs more introduction.

Results

1. Figure 1. Please correct the typo “Not RCT desigh (n=12)”.

2. Line 132-134. Please delete them.

3. Table 2. Please consider making separate tables for age groups or rearranging this table to make it easier to read.

4. Line 151: Please correct the typo. “… are presented in Figure 2”. Should be Table 2.

5. Tables 3-5. In my opinion, in this study, the left side of the forest plot should be labeled favors control, and the right side should be labeled favors experimental.

Discussion

1. Line 222. The authors mentioned that “The present meta-analysis demonstrated a small-to-moderte effect of acute endurance exercise on BDNF.” Please clarify small-to-moderate effect.

2. Line 267-270:” In addition, serum BDNF is more sensitive than plasma BDNF as a parameter in determining the effects of endurance exercise. Thus, serum BDNF should be used to assess the effects of acute endurance exercise on BDNF to avoid contradictory findings.” This is confusing. If it is already known that serum BDNF is more sensitive than plasma BDNF, it seems that in this study, there is no need to include the RCTs that reported plasma BDNF.

3. Line 272-273:” Previous studies suggested that exercise training promotes BDNF levels regardless of health status and age group.” Please add references here.

4. Line 275-276: “This finding provides insights for clinical and rehabilitation practitioners to design the prescription intervention of endurance exercise.” Please be more specific and discuss more future study directions.

Others

English requires improvement. Carefully checking to eliminate grammatical errors is needed.

Author Response

Comments

Correction

This meta-analysis reviewed RCTs that investigated the effect of acute endurance exercise on the level of brain-derived neurotrophic factor (BDNF). I have some comments.

Thank you for your review of our manuscript for the quality of our study and manuscript

1. Line 12: “Brain-derived neurotrophic factor plays an important role in …” Please add the abbreviation, BDNF, for brain-derived neurotrophic factor here.

We have added the abbreviation of BDNF in the first sentence.

2. Line 13-15: “Acute endurance exercise has been confirm that can effectively trigger increase of the expression of BDNF in peripheral blood. Therefore, investigating impact of acute endurance exercise on BDNF expression in peripheral blood is necessary.” These two sentences are confusing. Please rewrite them to make it clearer.

We have rewrite these two sentences to make it clear:

Whereas, acute endurance exercise is the key factor to trigger the expression of BDNF in peripheral blood. Investigating concentration of BDNF in peripheral blood can effectively assess impact of acute endurance exercise on brain health.

Introduction

1. Line 55-56:” Several studies have reported that acute endurance exercise significantly elevates the BDNF concentration in various brain areas.” Please add references here.

We have added the references of Seifert et al.(2010) and Erickson et al. (2011) here.

2. Please consider rewriting the introduction part. It is unclear to me why the authors focused on the effect of acute endurance exercise instead of other exercises on BDNF.

We have rewriting the introduction to make it clear why we focused on the effect of acute endurance exercise as well as add the description of plasma and serum BDNF and age groups .

3. In the discussion part, the authors mentioned the difference between plasma and serum BDNF. Please describe plasma and serum BDNF in the introduction part as well, since this study tested the effects of acute endurance exercise on plasma and serum BDNF separately.

4. The authors aimed to test the effect of acute endurance exercise in different age groups. This part also needs more introduction.

Results

1. Figure 1. Please correct the typo “Not RCT desigh (n=12)”.

We have corrected this typo from “Not RCT desigh” to “Not RCT design”.

2. Line 132-134. Please delete them.

We have deleted ”This section may be divided by subheadings. It should provide a concise and precise description of the experimental results, their interpretation, as well as the experimental conclusions that can be drawn.” in our manuscript.

3. Table 2. Please consider making separate tables for age groups or rearranging this table to make it easier to read.

We have rearranged Table 2 according to your suggestions.

4. Line 151: Please correct the typo. “… are presented in Figure 2”. Should be Table 2.

We have corrected this typo from “Figure 2” to “Table 2” in our manuscript.

5. Tables 3-5. In my opinion, in this study, the left side of the forest plot should be labeled favors control, and the right side should be labeled favors experimental.

This is a good suggestion to make results more clearly. We try to replot the forest plot in meta-analysis software (5.4.1 RevMan), have not figure out this question. We are sorry for that we can not figure out this question within 10 days.

Discussion

1. Line 222. The authors mentioned that “The present meta-analysis demonstrated a small-to-moderte effect of acute endurance exercise on BDNF.” Please clarify small-to-moderate effect.

According to an arbitrary but commonly used interpretation of effect size by Cohen (1988), a standardised mean effect size of 0 means no change, negative effect sizes mean a negative change, with 0.2 a small change, 0.5 a moderate change, and 0.8 a large charge. The SMD was 0.30, which is located in range from 0.2 to 0.5. Therefore, we describe this effect size as small to moderate.

2. Line 267-270:” In addition, serum BDNF is more sensitive than plasma BDNF as a parameter in determining the effects of endurance exercise. Thus, serum BDNF should be used to assess the effects of acute endurance exercise on BDNF to avoid contradictory findings.” This is confusing. If it is already known that serum BDNF is more sensitive than plasma BDNF, it seems that in this study, there is no need to include the RCTs that reported plasma BDNF.

BDNF in plasma and serum are two biomarker to representing BDNF concentration in central nervous system, which positively correlated with performance (Cho et al., 2012). Studies used BDNF both in plasma and serum to assess potential impact of exercise on brain function (Zoladz et al., 2011). For instance, Rodziewicz et al. (2020) used BDNF in plasma to investigate the effect of endurance exercise. Whereas, Arazi et al. (2021) used BDNF in serum to investigate the effect of endurance exercise. However, it is unclear if the difference of BDNF in plasma- or serum blood is better in assessing the effects of acute endurance on BDNF expression. In addition, Physiology study reported that there is a 200-fold difference exists between serum and plasma BDNF concentrations due to BNDF released mainly during coagulation.

Therefore, we take the subgroup analysis to investigate effect of acute endurance exercise on BDNF in plasma and serum. To clearly show why we analysis BDNF in plasma and serum, we have showed these information in introduction part (from line 77 to line 88).

3. Line 272-273:” Previous studies suggested that exercise training promotes BDNF levels regardless of health status and age group.” Please add references here.

We have added three reference here. They are  Marius et al. (2019) and Feter et al.(2019), respectively.

4. Line 275-276: “This finding provides insights for clinical and rehabilitation practitioners to design the prescription intervention of endurance exercise.” Please be more specific and discuss more future study directions.

We have further discussed impact of our meta-analysis finding in discussion part from line 307 to line 315 ; the detailed statement is as following:

It is well documented that physical exercise can stimuli BDNF expression and  increase BDNF concentration in human. This physical training-induced up-regulation of BDNF experssion plays a important role in human health, which has been confirmed that can improve functioning capacity in human, such as strength performance, endurance performance, mood and cognitive functions. Study even reported that effects of exercise on BDNF produce the similar effects of pharmacological treatment with antidepressant drug. Therefore, this findings from our study provides insights for clinical and rehabilitation practitioners to design the prescription of exercise intervention for neurological disorders.

Others

English requires improvement. Carefully checking to eliminate grammatical errors is needed.

We have submitted our manuscript into a native US speaker. She had carefully reedited our manuscript.  

Round 2

Reviewer 1 Report

To the attention of the authors,

The manuscript has improved considerably after revision and I understand the given responses.

I would also recommend to check spelling and punctuation before publishing the final version, paying special attention to the modified sections.

Reviewer 3 Report

I appreciate the authors' response. No further comments.